# Propranolol Reduces Epistaxis in Hereditary Hemorrhagic Telangiectasia: A Large Retrospective Study

**DOI:** 10.3390/jcm15010372

**Published:** 2026-01-04

**Authors:** Marcelo Martín Serra, Vanina Pagotto, Luisa Maria Botella, Carmelo Bernabeu

**Affiliations:** 1HHT Unit, Argentine Rendu Study Group (ARG), Internal Medicine Department, Hospital Italiano de Buenos Aires, Buenos Aires C1199ABB, Argentina; 2Argentine Rendu Study Group (ARG), Research Department, Hospital Italiano de Buenos Aires, Buenos Aires C1199ABB, Argentina; vanina.pagotto@hospitalitaliano.org.ar; 3Center for Biological Research “Margarita Salas”, Spanish National Research Council (CSIC), 28040 Madrid, Spain; cibluisa@cib.csic.es (L.M.B.); bernabeu.c@cib.csic.es (C.B.); 4Spanish National Network for Research on Rare Diseases (CIBERER), 28029 Madrid, Spain

**Keywords:** angiogenesis, adrenergic beta-antagonists, epistaxis, HHT, nasal hemorrhage

## Abstract

**Background/Objectives:** Hereditary Hemorrhagic Telangiectasia (HHT) is an autosomal dominant vascular dysplasia characterized by recurrent epistaxis, anemia, and visceral arteriovenous malformations. Epistaxis is the most frequent and disabling manifestation, with limited effective pharmacological options. Propranolol, a non-selective beta-blocker with vasoconstrictive and antiangiogenic properties, has shown benefit in other vascular anomalies but remains scarcely studied in HHT. This study aimed to evaluate the effect of oral propranolol on nasal bleeding in patients with HHT. **Methods:** A retrospective observational study including 151 adults with HHT (44 treated with propranolol, 107 untreated) was conducted using data from an Institutional HHT Registry from a referral center. Baseline demographic and clinical variables were recorded. Outcomes at 6 months included changes in hemoglobin, adherence to nasal hygiene, use of bleeding-related therapies, and improvement in epistaxis frequency and intensity according to the Sadick–Bergler scale. Logistic regression models were adjusted for confounders and indication bias using inverse probability of treatment weighting (IPTW). **Results:** After IPTW adjustment, propranolol was significantly associated with reduced frequency of epistaxis (adjusted OR: 3.8; 95% CI: 1.3–11.2; *p* = 0.016), while no effect was observed on intensity. Hemoglobin levels increased modestly in both groups without a significant difference. Patients without propranolol showed greater antifibrinolytic use, whereas adherence to nasal care remained stable among treated patients. **Conclusions:** Oral propranolol reduced nasal bleeding frequency in HHT, even among patients with greater baseline severity. Given its accessibility, safety, and potential to lessen treatment burden, it may represent a valuable adjunct therapy. This study represents the largest cohort of HHT patients treated with propranolol reported to date. Randomized trials including standardized bleeding scores and patient-reported outcomes are warranted to confirm clinical and quality-of-life benefits.

## 1. Introduction

Hereditary Hemorrhagic Telangiectasia (HHT), also known as Osler–Weber–Rendu disease, is an autosomal dominant vascular dysplasia [1,2] with an estimated prevalence of 1 in 5000 individuals [3,4]. It is caused by mutations in components of the BMP/ENG/ALK1/Smad4 signaling pathway, which play a key role in angiogenesis [5]. Clinically, HHT is characterized by telangiectasias affecting the skin and mucosal surfaces, particularly in the nose, oral and gastrointestinal tract, as well as by visceral arteriovenous malformations (AVMs) involving the central nervous system, lungs, liver, and gastrointestinal tract. Telangiectasias are structurally fragile, and their frequent rupture leads to typical manifestations such as recurrent epistaxis, gastrointestinal bleeding, and iron deficiency anemia [6]. AVMs, on the other hand, may result in organ-specific complications, including cerebral hemorrhage, thromboembolic or infectious events secondary to pulmonary AVMs (such as ischemic stroke or brain abscess), and high-output cardiac failure with secondary pulmonary hypertension or portal hypertension, due to hepatic vascular malformations (HVMs) [2]. The diagnosis of HHT is based on the Curaçao criteria and/or genetic testing [7,8].

Current treatment strategies focus on preventing complications related to vascular malformations and reducing nasal and gastrointestinal bleeding, while curative approaches remain unavailable [2,9]. Epistaxis is the most common clinical manifestation, affecting up to 95% of patients. Due to its severity and recurrence, it can severely impair quality of life, cause disability, or even become life-threatening [10,11]. Hygiene, lubrication, or moisturizing represent the first step of treatment of epistaxis in HHT. Although various ablative and surgical interventions (laser, sclerotherapy, electrocoagulation, septodermoplasty, nasal closure, etc.) are available, none have demonstrated complete efficacy, tolerability, or broad accessibility [9,12,13]. Pharmacological strategies, including antiangiogenic agents, aimed at reducing abnormal vessel formation, as well as antifibrinolytic therapies that stabilize clots and limit bleeding, represent an attractive and rational approach for more sustained and acceptable management of epistaxis [9,14,15].

Propranolol is a non-selective beta-blocker widely used for cardiovascular conditions and for other indications such as migraine, essential tremor, hyperthyroidism, and cancer [16]. Beyond these applications, it exerts well-documented antiangiogenic effects, being the first-line treatment for infantile hemangioma and designated as an orphan drug for other vascular anomalies [17,18,19]. Propranolol-mediated β-adrenergic receptor blockade induces vasoconstriction and reduces hypoxia-inducible factor 1 (HIF-1) nuclear translocation, leading to inhibition of pro-angiogenic target genes, including vascular endothelial growth factor (VEGF), matrix metalloproteinase 9 (MMP9), endoglin (ENG), and fibroblast growth factor (FGF). In addition, it decreases phosphorylation of the extracellular signal-regulated kinase/mitogen-activated protein kinase (ERK/MAPK) and proto-oncogene tyrosine protein kinase Src pathways, thereby activating caspases, promoting apoptosis, and ultimately suppressing angiogenesis [19,20,21]. Several small studies with limited quality of evidence have explored the systemic or topical use of propranolol and other beta-blockers for epistaxis in HHT, with encouraging results [19,21,22,23,24].

The aim of the present study was to evaluate the effect of propranolol on epistaxis in patients with HHT. Secondary objectives included comparing baseline and 6-month clinical characteristics, assessing adverse events, and describing the dosages used.

## 2. Materials and Methods

An observational, retrospective, and analytical study was conducted on HHT adult patients with epistaxis, using data from the Institutional HHT Registry of the HHT Unit at Hospital Italiano de Buenos Aires (Argentina), a national referral center. Although this study was also registered on ClinicalTrials.gov (NCT01761981), it is neither a retrospective analysis of a randomized controlled trial (RCT) nor an RCT itself. The study period extended from February 2010 through December 2019.

### 2.1. Patients

Adults with a confirmed diagnosis of HHT, based on clinical criteria (≥3 Curaçao criteria) and/or genetic confirmation, who were registered in the Institutional HHT Registry and presented with nasal bleeding (epistaxis), were included. Patients with incomplete or unavailable data in the registry were excluded.

### 2.2. Variables

The main explanatory variable was propranolol use. Baseline demographic and clinical data included sex, age, place of residence, nationality, health coverage insurance, death, and cause of death. These variables were collected to characterize the cohort and to assess potential differences in access to care and follow-up frequency (Table 1). The residency of excluded patients outside Buenos Aires was associated with a significantly higher likelihood of exclusion. Considering that propranolol treatment is off-label and its use is limited in those areas, patients living closer to our center may be more likely to receive this indication, as their proximity facilitates follow-up. Furthermore, patients from the city of Buenos Aires tend to attend consultations more frequently, which enables better monitoring and therapeutic adjustments.

Given the systemic nature of HHT, the presence of vascular malformations in different organs was also recorded and summarized in the baseline characteristics table, including gastrointestinal telangiectasias and other visceral arteriovenous malformations. In designing this study, we evaluated epistaxis using both the Epistaxis Severity Score (ESS) and the Sadick–Bergler scale, as these are the tools currently used at our center. Unfortunately, at the start of this study, there were not enough patients with complete data for both scales (Sadick–Bergler and ESS). By contrast, all patients had Sadick–Bergler assessments; therefore, we decided to use this scale for the present study. The primary outcome was improvement at 6 months, defined as a reduction of at least one point in the Sadick–Bergler scale [25] (Table 2) for either frequency or intensity of epistaxis. This variable was analyzed both as an ordinal outcome and as a dichotomous categorical variable, based on baseline Sadick–Bergler scores categorized as >3 versus ≤3.

Secondary outcomes included the following: changes from baseline to 6 months in hemoglobin levels; adherence to nasal hygiene and lubrication; use of bleeding-modifying treatments (pharmacological or surgical); iron requirements (oral and/or intravenous); use of antifibrinolytic and antiangiogenic agents (including bevacizumab); and overall indicators of bleeding burden such as iron metabolism parameters, iron supplementation, red blood cell transfusions, and anemia.

For propranolol-treated patients, other treatment indications were recorded. When a classical indication was absent, the prescription was categorized as *expert judgment* (not analyzed as a separate variable). Adverse events related to propranolol and prescribed doses were also assessed as categorical variables.

### 2.3. Sample Size and Statistical Analysis

Because the effect of propranolol on HHT-related epistaxis has received limited investigation, and given the retrospective and observational design of this study, all eligible patients in the registry were included. For the descriptive analysis, quantitative variables were expressed as mean ± standard deviation or as median and interquartile range (25–75%), according to distribution. Qualitative variables were expressed as absolute and relative frequencies. Comparisons between included and excluded groups, as well as between propranolol and non-propranolol groups, were performed using chi-square or Fisher’s exact test for categorical variables, and Student’s *t*-test or Mann–Whitney test for continuous variables, according to distribution.

Pre–post comparisons (baseline vs. 6 months) were conducted within each treatment group. For continuous variables such as hemoglobin, paired *t*-tests or Wilcoxon signed-rank tests were applied, depending on data distribution. For categorical variables, McNemar’s test was used when two response categories were present, and the Stuart–Maxwell test when variables had more than two categories. Due to the observational character of this study, the non-random assignment of drug treatment (propranolol) may introduce biases, since there are systematic differences between the groups compared. To control these differences and to estimate more precisely the effect of drug treatment, the IPTW was used [26]. The propensity scores were estimated for each participant, and the overlap of covariates between treatment groups was assessed using a common support plot. A Love Plot was used to assess the effectiveness of IPTW in balancing covariates between treatment groups.

Subsequently, inverse probability weights of treatment were calculated for each individual. Finally, these weights were used in a multiple logistic regression to estimate the effect of propranolol on bleeding improvement at 6 months after its initiation. The adjusted covariates included age, oral and/or iv iron at baseline, minor surgery at baseline, compliance with nasal care measures, other pharmacological treatment besides propranolol, Sadick–Bergler baseline frequency ≥ 3, and the propensity score estimated through IPTW. The same analysis was performed separately for each bleeding outcome: intensity and frequency. Differences were considered statistically significant at *p* < 0.05. All statistical analyses were performed with R software version 4.3.3.

## 3. Results

As of 31 December 2019, the Institutional HHT Registry included 510 patients, of whom 151 were eligible and included in this analysis, according to the criteria outlined in Figure 1.

To evaluate whether the exclusion of 201 patients could have introduced a selection bias (e.g., loss to follow-up, death, or other factors that might compromise the internal validity of the study), we compared the baseline characteristics of excluded versus included patients. The included patients were older and had a higher prevalence of AVMs. No significant differences were observed in bleeding severity as assessed by the Sadick–Bergler scale. These comparisons are shown in Table 3.

Among the 151 patients analyzed, 56 (37.1%) were male, and the median age was 50 years (IQR 37–65). This study reflects a predominance of women in both cohorts, a finding consistent with what has been observed in many studies on HHT, including our own prevalence study. Although HHT is an autosomal dominant disease, we believe this difference could be due, at least in part, to consultation bias, as women tend to have greater access to healthcare systems. In addition, women tend to have milder epistaxis than men before menopause, likely due to the protective effect of female hormones. However, after menopause, epistaxis severity increases in women as hormone levels decline. Since most of the women included in our cohort were beyond reproductive age, this hormonal shift may also reasonably explain the female predominance observed. The propranolol group had a higher median age compared with the control group (62 vs. 47 years). This difference is probably explained by the age-dependent progression of epistaxis in HHT, with older patients being more likely to seek medical care.

Propranolol was prescribed in 44 patients (29.1%), of whom 33 (75%) had an additional clinical indication: migraine in 9 (20.5%), hypertension in 14 (31.8%), and tachyarrhythmias or palpitations in 10 (22.7%). In 11 patients (25%), propranolol was prescribed with the primary intention of treating epistaxis, considered an expert-driven decision without a conventional cardiovascular indication. In this subgroup, the rationale for prescription was based on the anticipated antiangiogenic and vasoconstrictive properties of propranolol, described herein as an expert indication for treatment with the intention to control bleeding.

Adverse events were reported in six patients: hypotension in three, sleep disturbances in one, and mood alterations in two. Patients in the propranolol group were older and more frequently received additional pharmacological therapies (other than propranolol), particularly antifibrinolytics. Table 4 summarizes the baseline characteristics of patients according to propranolol prescription. In terms of nasal lubrication and hygiene, adherence was higher among patients treated with propranolol, who showed better compliance and lower rates of non-adherence. We also observed an increase in nasal lubrication and humidification in the non-propranolol group. This is likely attributable to the supportive counseling, education, and positive reinforcement provided during follow-up consultations. No significant differences were observed between groups regarding the presence of vascular malformations in other organs, surgical procedures, or conditions such as anticoagulant use or additional hemorrhagic disorders unrelated to HHT. Similarly, bleeding severity, as measured by the Sadick–Bergler scale, was comparable across groups both for individual domains and for the dichotomized evaluation (≥3 vs. <3) of intensity and frequency.

### 3.1. Results for Propranolol Dosage

Regarding prescribed doses, 35 patients (79%) received low doses (20–60 mg), while eight patients (18%) received intermediate doses (80–160 mg). No patients received high doses (>240 mg). No significant differences were observed between the groups (*p* = 0.597).

### 3.2. Results for Gastrointestinal Bleeding

Gastrointestinal (GI) bleeding was assessed in patients aged >50 years, both with and without propranolol treatment. This measure was applied to minimize selection bias, as after this age, patients are more likely to undergo gastrointestinal evaluation either through colorectal cancer screening protocols (which in our center almost always include upper endoscopy) or because GI bleeding due to HHT becomes more frequent beyond this age. Thus, the study population included 48 patients corresponding to the non-propranolol group and 32 for the propranolol-treated group. Among the 48 patients not treated with propranolol, 9 (18.8%) presented with GI bleeding at 6 months, compared with 8 out of 32 patients (25%) in the propranolol group. At baseline, only one patient in the non-propranolol group presented bleeding, whereas in the propranolol group, two patients who had no bleeding at baseline developed bleeding during follow-up. No significant differences were observed between groups (*p* = 0.535).

### 3.3. Hemoglobin and Iron Supplementation

Changes in hemoglobin levels before and after 6 months were assessed in both propranolol-treated and untreated patients. No significant differences were observed between baseline and 6-month values in either group (Figure 2). Patients treated with propranolol, however, had lower baseline hemoglobin levels, indicating more severe anemia (Table 4).

Additionally, changes in iron supplementation from baseline to six months were analyzed in both groups. No significant within-group differences were observed over time in intravenous iron use (without propranolol, *p* = 0.79; with propranolol, *p* = 1.00), indicating stable utilization. Similarly, no significant changes were found in oral iron use (without propranolol, *p* = 0.42; with propranolol, *p* = 1.00), suggesting consistent oral supplementation throughout follow-up (Figure 3).

### 3.4. Nasal Care and Lubrication/Moisturizing

In the non-propranolol group, the distribution of adherence categories changed significantly between baseline and 6 months *p* < 0.001), showing a decrease in the proportion of patients with insufficient adherence and an increase in those with partial and optimal adherence. In contrast, no significant differences were observed in the propranolol group (*p* = 0.777). Figure 4 shows the distribution of hygienic–dietary adherence categories at baseline and at 6-month follow-up according to propranolol treatment.

### 3.5. Pharmacological Treatments Other than Propranolol

The proportion of patients receiving antiangiogenic therapy remained unchanged between baseline and 6-month follow-up, with no significant differences in either group (without propranolol, *p* = 1.00; with propranolol, *p* = 1.00). A significant increase in antifibrinolytic use was observed among patients without propranolol (*p* = 0.007), whereas no significant change was detected in those treated with propranolol (*p* = 1.00)

The proportion of patients receiving bevacizumab therapy remained unchanged between baseline and 6-month follow-up. With no significant differences in either group (without propranolol, *p* = 1.00; with propranolol, *p* = 1.00). Figure 5 shows the distribution of pharmacological treatments other than propranolol at baseline and at 6-month follow-up according to propranolol treatment.

### 3.6. Association Between Propranolol Use and Improvement in Epistaxis

Both groups showed a significant shift toward lower category Sadick–Bergler intensity scores at follow-up (with propranolol, *p* < 0.001; without propranolol, *p* < 0.001), indicating overall improvement (Figure 6).

When changes in bleeding frequency according to the Sadick–Bergler scale were analyzed (Figure 7), both groups showed a redistribution toward lower categories, including an increased proportion of patients in category 0 (no bleeding). This improvement in epistaxis was observed in both groups: in the untreated group (*p* = 0.002), it was likely attributable to better nasal care, whereas in the propranolol group (*p* = 0.001), it reflected the combined effect of nasal care and propranolol. However, the reduction in bleeding frequency was more pronounced in the propranolol-treated group, where a clear increase in the proportion of patients in categories 0 and 1 was noted, accompanied by a corresponding decrease in categories 2 and 3 (Figure 7).

For further analysis, the Sadick–Bergler scale was dichotomized into <3 (milder cases) versus ≥ 3 (severe bleeding). Regarding intensity, the proportion of patients with Sadick–Bergler intensity ≥ 3 significantly decreased in both groups (with propranolol, *p* = 0.023; without propranolol, *p* = 0.014), indicating a reduction in the proportion of patients with higher-intensity epistaxis at 6 months (Figure 8).

Regarding frequency, the proportion of patients with Sadick–Bergler frequency ≥ 3 significantly decreased in the propranolol group (*p* = 0.044), while no significant change was observed in the non-propranolol group (*p* = 0.44), indicating a reduction in high-frequency epistaxis only among treated patients (Figure 9).

### 3.7. Adjustment for Indication Bias

To account for potential indication bias, IPTW based on the propensity score was applied. Figure 10 shows the distribution of estimated propensity scores for patients treated and not treated with propranolol, illustrating the region of common support between groups. Figure 11 presents the standardized mean differences (SMDs) for baseline covariates before and after weighting. After IPTW adjustment, all covariates achieved adequate balance (SMD < 0.1), indicating successful reduction of baseline differences between groups.

Improvement in epistaxis, defined as a reduction of at least one point in any category of the Sadick–Bergler scale, was observed in 34 of 44 patients treated with propranolol (77%) compared with 63 of 107 patients without propranolol (58.9%). No significant association was found between propranolol use and overall improvement in bleeding, either in the unadjusted analysis or after adjusting for indication bias using IPTW (Table 5).

When outcomes were analyzed separately, 26 of 44 patients treated with propranolol (59.1%) showed improvement in bleeding intensity compared with 53 of 107 patients in the non-propranolol group (49.5%). Regarding frequency, improvement was observed in 27 of 44 patients treated with propranolol (61.4%) versus 38 of 107 patients without propranolol (35.5%). Stratified analysis of the Sadick–Bergler scale categories confirmed a significant association between propranolol use and improvement in frequency, both in crude and adjusted analyses, whereas no association was found for intensity (Table 6).

## 4. Discussion

This study suggests that propranolol is useful for the treatment of epistaxis in HHT patients, particularly through the reduction of bleeding frequency and probably also intensity. Therapeutic strategies in HHT remain limited. Although in recent years, relevant advances have been made in understanding its pathophysiological mechanisms, current management is mainly based on the repurposing of drugs approved for other indications that can modulate biological processes involved in the disease. Some of these include hormonal analogs and antiangiogenic agents, and some are designated as orphan drugs in Europe, such as raloxifene and bazedoxifene [27,28]. Propranolol has emerged as a promising candidate, although clinical evidence is still limited due to heterogeneity in its pharmacological administration, low statistical power, and study design [19,27,29]. The rationale for propranolol repurposing is based on multiple biological properties: antiangiogenic action, modulation of signaling pathways, vasoconstrictive effect, and usefulness in other common HHT clinical manifestations (migraines, arrhythmias, portal hypertension) or comorbidities such as arterial hypertension. Propranolol is a safe, inexpensive, and widely used drug, although its beta-blocking effect could compromise compensatory mechanisms in severe hemorrhage, which explains the preference of some groups for topical alternatives such as timolol [29]. However, experience in infantile hemangioma suggests that antiangiogenic effects are mainly achieved with systemic administration [18].

In this study, 151 patients were included from a total of 510 initially evaluated. Only significant differences in age and visceral malformations were observed, both of which were more frequent in the included patients compared to the excluded individuals. This suggests a study population with greater clinical severity, more comorbidities, and closer contact with the healthcare system, making the results generalizable to older and more complex HHT patients. Indeed, the included patients were significantly older and had a higher number of AVMs than the excluded population. These differences can be explained by the study design, as patients without a definite HHT diagnosis were excluded. Given the age-related penetrance of HHT, older individuals are more likely to fulfill a greater number of diagnostic criteria. Among those patients treated with propranolol, characteristics of greater severity, higher drug use (particularly antifibrinolytics), and better adherence to nasal care measures were observed, likely reflecting greater interaction with HHT specialists. These baseline differences are expected in an observational study, unlike randomized clinical trials with propranolol or topical timolol, in which no significant discrepancies between groups were found [23,30,31]. Regarding gastrointestinal bleeding, no baseline differences were observed between groups, although the analysis was limited by the absence of endoscopic studies in several patients, particularly those under 50 years of age.

To the best of our knowledge, no clinical trial has evaluated the efficacy of propranolol in this context, whereas studies on gastrointestinal bleeding in HHT are limited, probably due to the complexity of its assessment. Differences in the therapeutic response to other drugs have also been observed: tranexamic acid has not shown efficacy in this scenario [13], while pazopanib or bevacizumab appear more effective for gastrointestinal than for nasal bleeding [32,33,34], likely reflecting variations in the extent, localization, and biological microenvironment of telangiectasias.

Hemoglobin levels and their variations were considered as indirect indicators of bleeding. Since hemoglobin depends largely on iron levels, and these on the balance between intake and losses, an additional variable was included to account for variations in iron supplementation, both oral and intravenous, though dietary iron intake could not be considered. No statistically significant variations in hemoglobin levels were observed in either group, despite a noticeable increase in propranolol-treated patients. Notably, patients in the propranolol group started with lower hemoglobin levels compared with the other group. This may indirectly indicate that they were more severe bleeders with deeper iron deficits.

Within-group comparisons between baseline and 6-month evaluations provided additional insight into the clinical evolution of patients. Iron supplementation requirements remained stable over time, whereas antifibrinolytic use increased significantly among patients without propranolol, indicating a need for additional pharmacological support in that group. Adherence to nasal care showed improvement among non-treated patients and stability among propranolol users. Finally, when analyzing the Sadick–Bergler scale, both groups exhibited a significant reduction in intensity scores, while only patients treated with propranolol showed a significant decrease in high-frequency epistaxis. Altogether, these trends highlight that while general supportive measures benefit all patients, propranolol appears to specifically reduce the recurrence of bleeding episodes.

From a clinical perspective, patients with greater compromise are generally those requiring more treatments. Conversely, those experiencing less frequent and/or less intense bleeding tend to respond better to iron supplementation, showing an increase in hemoglobin. In the study by Ichimura et al. (2016) with topical timolol in patients undergoing septodermoplasty, stabilization of hemoglobin with cessation of transfusion requirements was reported [35]. Using topical timolol in patients treated with laser, this was not observed in the TIM-HHT trial, in which no changes in hemoglobin or other iron-related parameters were found with the intervention [30]. In a pilot retrospective study with propranolol gel, the authors reported a significant increase in hemoglobin levels after 12 weeks of treatment in the intervention group, although this was not associated with a significant reduction in transfusion requirements [36]. These findings were confirmed in a subsequent trial conducted by the same group with 1.5% propranolol gel in a larger cohort over 8 weeks [31]. Dupuis-Girod and colleagues did not observe modifications in hemoglobin levels in the randomized double-blind trial of topical timolol versus placebo [37], either. A recent small trial with oral propranolol found a significant increase in hemoglobin after 3 months of treatment but not after six months [24]. Hemoglobin improvement in treated patients is variable across studies. Although study designs and pharmacological formulations differ, hemoglobin modifications depend on multiple factors. Other drugs used in HHT, such as tranexamic acid or thalidomide, although significantly reducing bleeding, have not shown consistent hemoglobin increases, except for bevacizumab, in which EPO gene involvement may also play a role.

We also assessed changes in oral and/or intravenous iron supplementation at baseline and after six months. In line with hemoglobin levels, a slightly higher proportion of patients in the propranolol group had increased iron requirements. Although intravenous iron often indicates more severe disease, it may also reflect intolerance to oral formulations, patient preference, accessibility, or cost. In HHT, where bleeding is chronic and unpredictable, iron therapy is rarely discontinued.

Regarding hygienic and lubrication measures, nasal care is the first-line therapy for epistaxis and strongly recommended in the Second International Guidelines for HHT [13]. In our center, patients are instructed in writing on proper technique and products. Adherence was inconsistent overall, though higher among those not receiving propranolol. Patients treated with propranolol showed stable adherence, possibly reflecting greater disease severity, more frequent clinical follow-up, or reinforcement of recommendations. No previous studies of beta-blockers in HHT have assessed changes in nasal care. Importantly, some topical trials suggest potential benefits from the vehicle itself, such as saline or thermosensitive gels [38].

Pharmacological treatments in HHT aim to compensate for haploinsufficiency, modulate angiogenesis, reduce inflammation, or optimize coagulation. They are often used in combination, as monotherapy rarely achieves control. We assessed changes in non-propranolol therapies during follow-up, grouping them into antifibrinolytics, antiangiogenics, and bevacizumab (considered separately given its stronger evidence). Fewer patients in the propranolol group required additional pharmacological treatment, suggesting that propranolol may reduce the need for other drugs to control epistaxis, although discontinuation due to adverse effects, lack of efficacy, accessibility, or patient preference may also account for this finding. Antifibrinolytic use increased only in the non-propranolol group, suggesting a synergistic role of propranolol with tranexamic acid. Consistent with prior studies [39], tranexamic acid was effective mainly in reducing severity rather than the frequency of bleeding, and in our practice, it remains the first-line agent after nasal care.

Regarding the primary outcome, propranolol was not significantly associated with overall improvement (frequency + intensity). However, patients receiving propranolol had an almost four-fold higher odds of improvement in frequency, with the lower CI close to 1 (OR: ~4, 95%; CI: 0.9–5.6) [40]. This trend aligns with the TIM-HHT trial, where timolol did not achieve significance but showed borderline improvements in bleeding scores [41]. Of note, our follow-up of 6 months may capture more durable clinical benefit than the shorter periods assessed in topical studies.

Trials with topical propranolol gel in patients with moderate epistaxis (ESS ≥ 4) demonstrated significant reductions in bleeding, transfusion requirements, and hemoglobin improvement, although nasal inspection did not differ [31,36]. Similarly, our data suggest propranolol is particularly effective in more severe cases. Other trials with topical timolol reported improvements but failed to demonstrate superiority over placebo, while systemic propranolol has shown beneficial effects in both cardiovascular/neurological indications and bleeding parameters [19]. A small trial compared oral propranolol (10 patients) versus placebo (5 patients). The authors did not observe morphological differences in the nasal telangiectasias, although they were able to estimate a reduction in epistaxis severity (ESS) in the propranolol arm after 3 months of treatment. However, the study presents notable limitations in both methodology and patient recruitment [24].

Mechanistically, propranolol may reduce shear stress, stabilize telangiectasias, and exert vasoconstrictive and antiangiogenic effects, thereby lowering the number of fragile lesions [42,43]. Its catecholamine-blocking action may also decrease stress-induced epistaxis, the most frequent trigger in our series. Conversely, PAI-1 inhibition may explain the limited effect on intensity, highlighting a complementary role for tranexamic acid [44].

In our study, most patients received low propranolol doses, precluding dose–response analysis. Although preclinical data support dose-dependency, our results suggest that even low doses may reduce bleeding frequency. A small randomized trial with oral propranolol reports benefits with 80 mg per day [24]. No standardized dosing exists for HHT-related epistaxis; extrapolation from infantile hemangioma (2–3 mg/kg/day) may be reasonable, though further trials are required [45].

Adverse effects were reported in six patients (hypotension, insomnia, or mood changes). As with other beta-blocker studies, events were mild, and none were severe [24,37]. While topical timolol has been associated with systemic absorption and hemodynamic effects [38,46], propranolol was generally well tolerated. Thus, systemic propranolol appears safer and more justified than topical formulations.

This study has limitations: First, there were fewer patients than expected, although baseline differences were limited to age and visceral AVMs, without affecting baseline epistaxis severity. We also faced the inability to apply ESS in most patients, requiring dichotomization of the Sadick–Bergler scale, which reduced statistical power but enabled separate evaluation of bleeding frequency and intensity. There were limited genetic data, precluding genotype-specific analyses. Furthermore, there was insufficient information to assess gastrointestinal bleeding response, and there was a lack of detailed dose–response analysis. The strengths of this study include that it represents the largest cohort reported to date, overcoming the methodological limitations of the few prior studies, which lacked placebo or untreated control groups or had a very small number of patients included. Although observational studies provide a lower level of evidence compared with randomized clinical trials, a major strength of our work is its real-world design, where follow-up frequency, treatment adherence, and patient behavior were not controlled, better reflecting clinical practice. The 6-month follow-up allowed evaluation of both the initial response and its maintenance. Analyses were adjusted for relevant clinical variables and indication bias using IPTW.

## 5. Conclusions

Oral propranolol is a low-cost and widely accessible drug with vasoconstrictive and antiangiogenic properties that effectively reduces the frequency of epistaxis in patients with HHT. Owing to its favorable safety profile, it may represent a valuable therapeutic option, particularly for older or comorbid patients who often have limited alternatives. Future randomized clinical trials that include a large cohort of patients and incorporate standardized bleeding scores, different propranolol doses, and patient-reported outcomes are warranted to confirm these findings and to better quantify the clinical benefit and quality-of-life improvements associated with propranolol therapy.

## Figures and Tables

**Figure 1 jcm-15-00372-f001:**
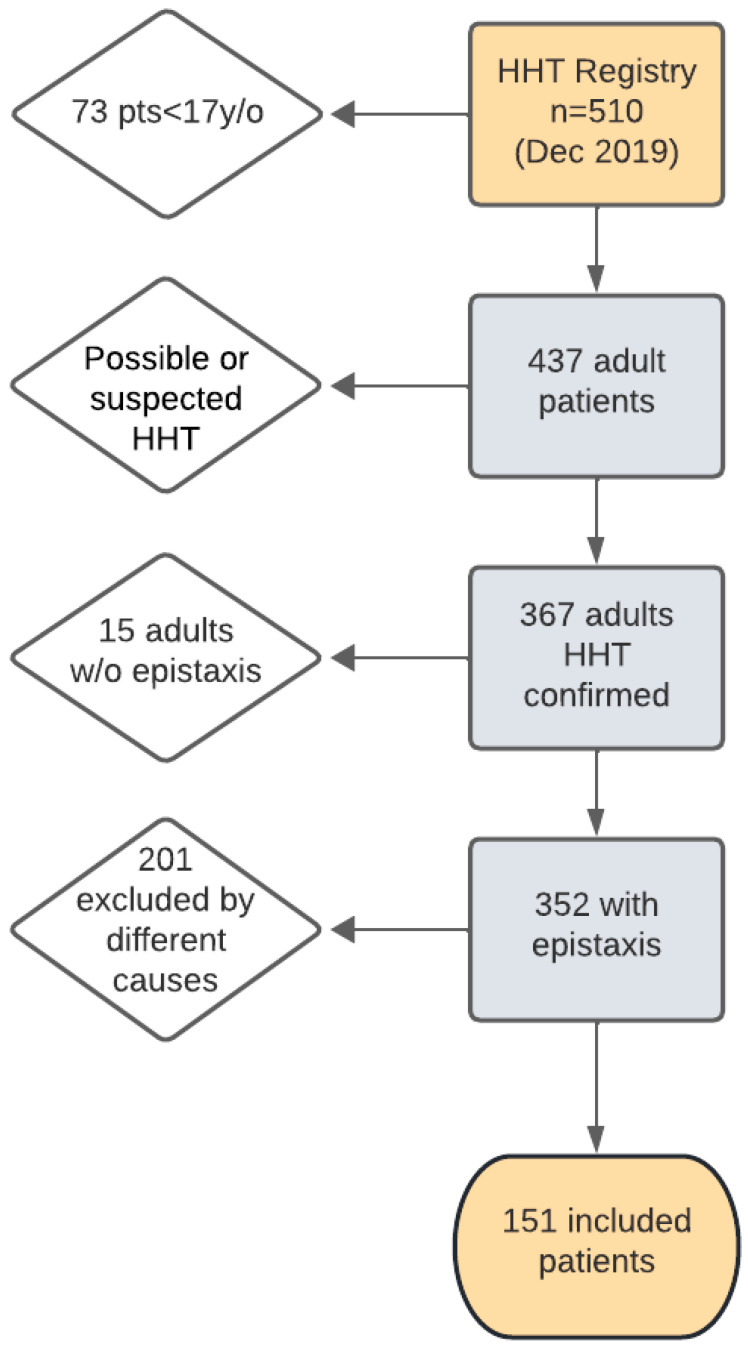
Patient flowchart patient selection from the Hereditary Hemorrhagic Telangiectasia (HHT) Registry (December 2019). Among the initial number of patients in the registry (510), the applied exclusion criteria were (i) being younger than 17y/o; (ii) having only suspected HHT diagnosis; (iii) absence of epistaxis; or (iv) additional causes, including lack of complete laboratory tests (45), only one single Sadick–Bergler measurement (25), two widely separated Sadick–Bergler measurements (79), unclassified epistaxis severity (23), laboratory data collected remote from the epistaxis event (22), deceased after a single follow-up visit (6), and withdrawal of consent (1). After the screening process, 151 patients were included in this study. pts, patients; y/o, years old; w/o, without.

**Figure 2 jcm-15-00372-f002:**
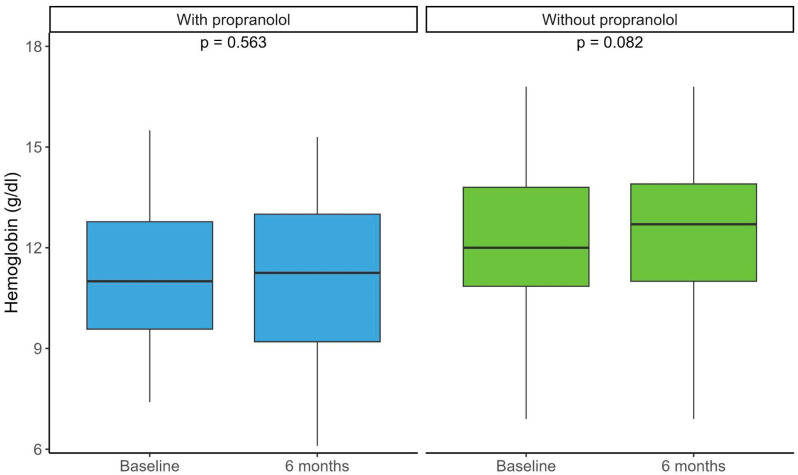
Distribution of hemoglobin levels (g/dL) in patients treated with and without propranolol at baseline and after 6 months of follow-up. Boxplots show the median (central line), interquartile range (box), and extreme values (whiskers). Within-group comparisons between time points were performed using paired tests (*p*-values indicated in the graph). With propranolol, the baseline median was 11 (RIC 9.5–12.8), and the 6-month median was 12.2 (RIC 9.2–13). Without propranolol, the baseline median was 12 (RIC 10.8–13.8), and the 6-month median was 12.7 (RIC 11–13.9). RIC, Relative Index Comparison.

**Figure 3 jcm-15-00372-f003:**
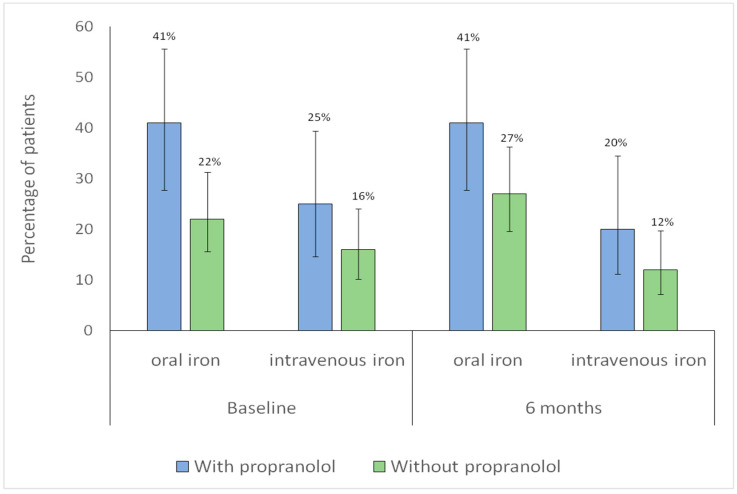
Distribution of iron supplementation at baseline and at 6-month follow-up according to propranolol treatment. Bars represent the proportion of patients receiving each type of iron supplementation in each group (blue: with propranolol; green: without propranolol). Vertical lines denote 95% confidence intervals. The numbers displayed above each bar correspond to the percentage of patients.

**Figure 4 jcm-15-00372-f004:**
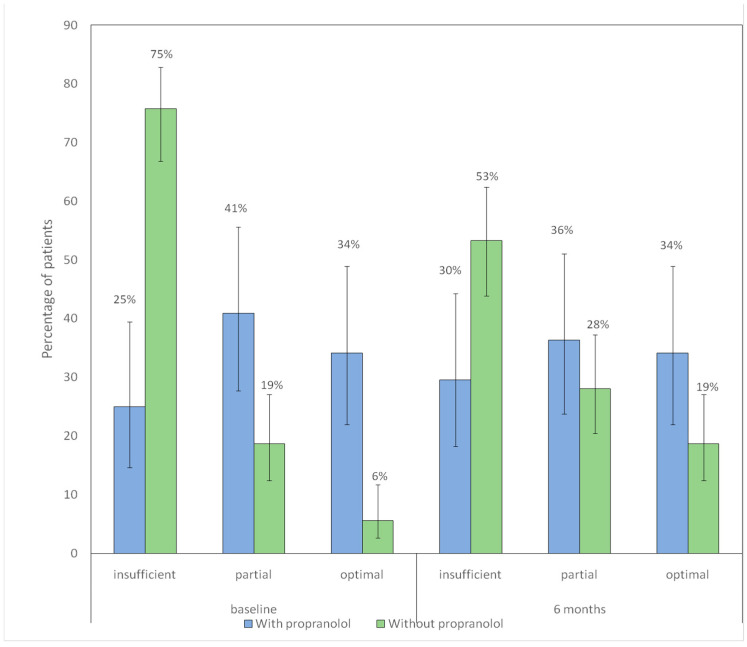
Distribution of hygienic–dietary adherence categories at baseline and at 6-month follow-up according to propranolol treatment. Bars represent the proportion of patients receiving each type of iron supplementation in each group (blue: with propranolol; green: without propranolol). Vertical lines denote 95% confidence intervals. The numbers displayed above each bar correspond to the percentage of patients.

**Figure 5 jcm-15-00372-f005:**
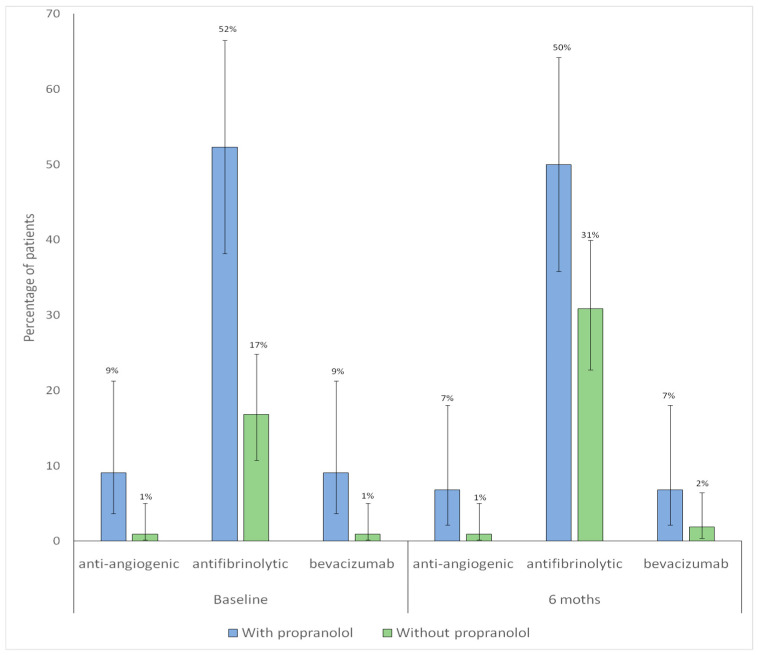
Distribution of pharmacological treatments (antiangiogenic, antifibrinolytic, and bevacizumab) other than propranolol at baseline and at 6-month follow-up according to propranolol treatment. Bars represent the proportion of patients receiving antiangiogenic, antifibrinolytic, or bevacizumab therapy in each group (blue: with propranolol; green: without propranolol). Vertical lines indicate 95% confidence intervals. The numbers displayed above each bar correspond to the percentage of patients.

**Figure 6 jcm-15-00372-f006:**
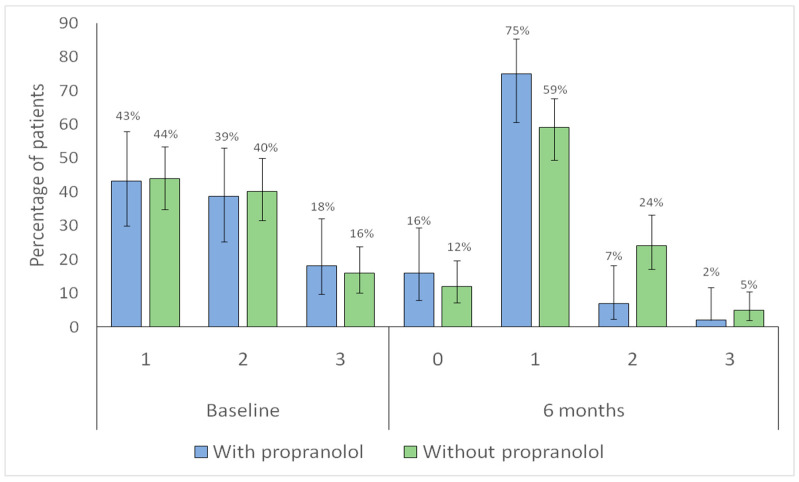
Distribution of Sadick–Bergler intensity scores categories at baseline and at 6-month follow-up according to propranolol treatment. Bars represent the proportion of patients within each Sadick–Bergler intensity category at baseline (1–3) and at 6-month follow-up (0–3), according to propranolol treatment (blue: with propranolol; green: without propranolol). Vertical lines indicate 95% confidence intervals. Numbers above each bar correspond to the percentage of patients in each category.

**Figure 7 jcm-15-00372-f007:**
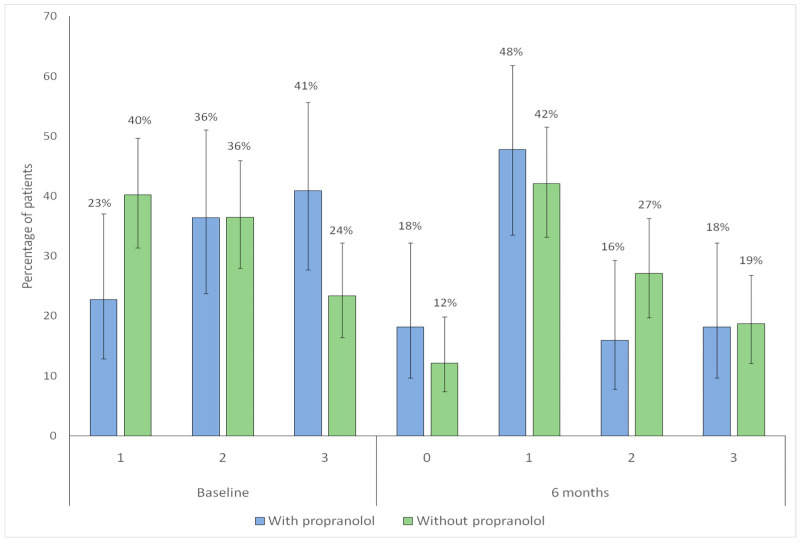
Distribution of Sadick–Bergler frequency scores categories at baseline and at 6-month follow-up according to propranolol treatment. Bars represent the proportion of patients within each Sadick–Bergler frequency category at baseline (1–3) and at 6-month follow-up (0–3), according to propranolol treatment (blue: with propranolol; green: without propranolol). Vertical lines indicate 95% confidence intervals. Numbers above each bar correspond to the percentage of patients in each category.

**Figure 8 jcm-15-00372-f008:**
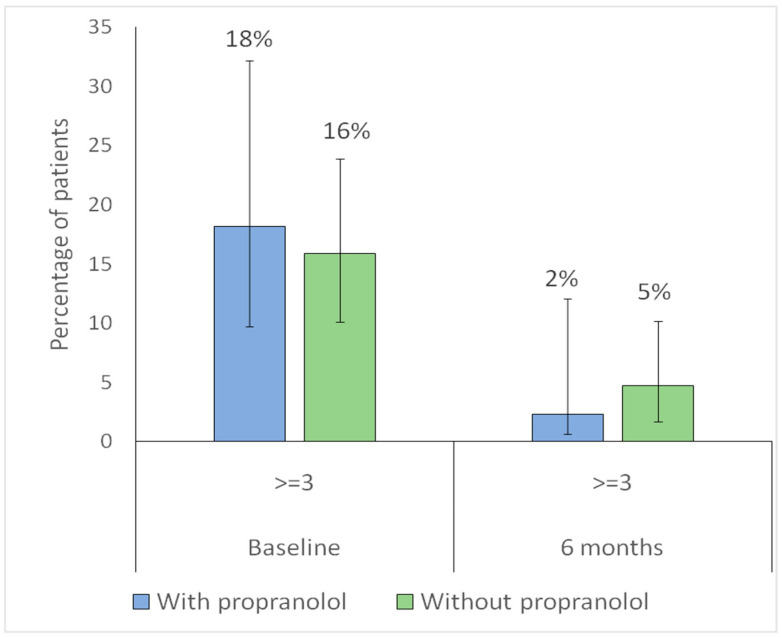
Distribution of Sadick–Bergler intensity scores ≥ 3 versus < 3 at baseline and at 6-month follow-up according to propranolol treatment. Bars represent the proportion of patients with Sadick–Bergler intensity score ≥ 3 at baseline and at 6-month follow-up, according to propranolol treatment (blue: with propranolol; green: without propranolol). Vertical lines indicate 95% confidence intervals. Numbers above each bar correspond to the percentage of patients in each group.

**Figure 9 jcm-15-00372-f009:**
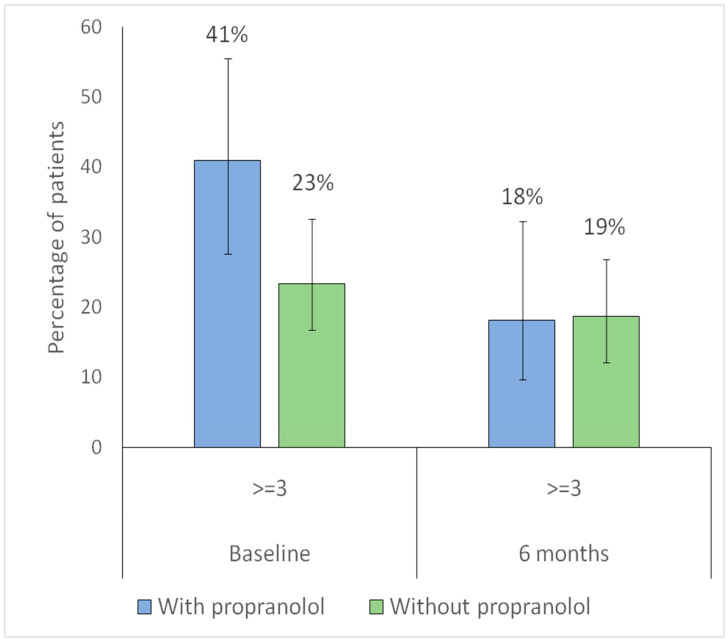
Distribution of Sadick–Bergler frequency scores ≥ 3 versus < 3 at baseline and at 6-month follow-up according to propranolol treatment. Bars represent the proportion of patients with Sadick–Bergler frequency score ≥ 3 at baseline and at 6-month follow-up, according to propranolol treatment (blue: with propranolol; green: without propranolol). Vertical lines indicate 95% confidence intervals. Numbers above each bar correspond to the percentage of patients in each group.

**Figure 10 jcm-15-00372-f010:**
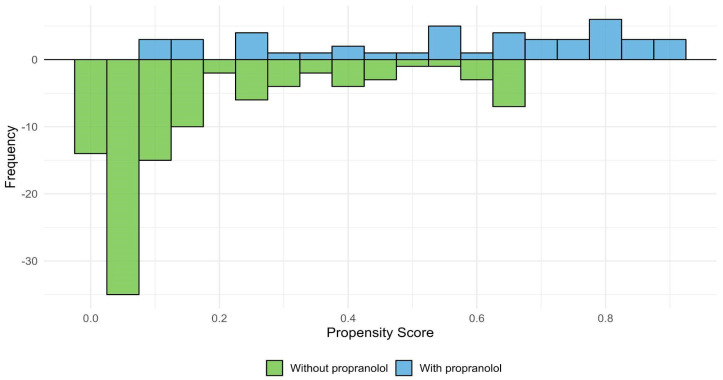
Propensity score distribution and region of common support. Distribution of estimated propensity scores for patients treated with propranolol (blue) and without propranolol (green). The *x*-axis represents the estimated propensity score for each individual, and the *y*-axis represents the frequency of patients within each group. The overlapping area indicates the region of common support, where both groups share comparable probabilities of receiving propranolol.

**Figure 11 jcm-15-00372-f011:**
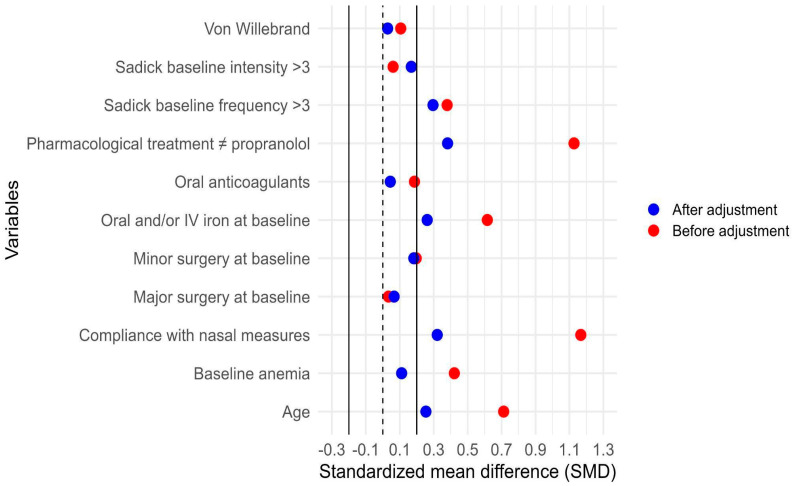
Covariate balance before and after inverse probability of treatment weighting (IPTW). Standardized mean differences (SMDs) for baseline covariates before (red) and after (blue) IPTW adjustment are shown. The vertical solid and dashed lines at 0 and ±0.1 represent thresholds for acceptable covariate balance. After weighting, all covariates retained in the final outcome model achieved SMD ± 0.1, indicating adequate balance between propranolol and non-propranolol groups. Variables outside the SMD threshold were not used to construct the IPTW but were included as additional adjustment variables when estimating the association between propranolol and the outcome.

**Table 1 jcm-15-00372-t001:** Gender, age, and geographical distribution of participants *.

	Excluded n = 201	Included n = 151	*p*-Value
**Gender n (%)**			0.902
Male	77 (38.3)	56 (37.1)	
Female	124 (61.7)	95 (62.9)	
Median Age (IQR)	43 (34–58)	50 (38.5–64)	0.003
**Place of origin n (%)**			0.074
Buenos Aires City	61 (30.3)	61 (40.4)	
Greater Buenos Aires	49 (24.4)	42 (27.8)	
Province of Buenos Aires	32 (15.9)	16 (10.6)	
Other Provinces	59 (29.4)	32 (21.2)	
**Residency outside of the metropolitan area of Buenos Aires, n (%)**			0.014
Yes	91 (45.3)	48 (31.8)	
No	110 (54.7)	103 (68.2)	

* The residence of excluded patients outside the metropolitan area of Buenos Aires was associated with a significantly higher likelihood of exclusion. Considering that propranolol treatment is off-label and its use is limited in those areas, patients living closer to our center may be more likely to receive this indication, as their proximity facilitates follow-up. Furthermore, patients from the city of Buenos Aires tend to attend consultations more frequently, which enables better monitoring and therapeutic adjustments.

**Table 2 jcm-15-00372-t002:** Sadick–Bergler scale.

Grade of Sadick–Bergler	Frequency of Bleeding	Intensity of Bleeding
Grade 1	Less than once per week	Slight stains on handkerchief
Grade 2	Several times per week	Soaked handkerchief
Grade 3	More than once per day	Bowl or equivalent container necessary

**Table 3 jcm-15-00372-t003:** Baseline characteristics of excluded and included patients.

	Excluded (n = 201)	Included (n = 151)	*p* Value
**Gender n (%)**			0.902
Female, n (%)	124 (61.7)	95 (62.9)	
Male, n (%)	77 (38.3)	56 (37.1)	
Median age (IQR)	43 (34–58)	50 (38.5–64)	0.003
AVMs, n (%)	106 (52.7)	124 (82.1)	<0.001
**Sadick–Bergler scale** **Intensity * n (%)**			
1	115 (57.2)	84 (55.6)	
2	63 (31.3)	50 (33.1)	0.939
3	23 (11.4)	17 (11.3)	
**Sadick–Bergler scale** **Frequency * n (%)**			
1	102 (50.7)	70 (46.4)	
2	46 (22.9)	25 (29.8)	0.341
3	53 (26.4)	36 (23.8)	
**Sadick–Bergler scale ****			
Intensity ≥ 3	53 (26.4)	36 (23.8)	0.677
Frequency ≥ 3	23 (11.4)	17 (11.3)	0.999

n, absolute frequency; IQR, interquartile range (25–75); AVMs, arteriovenous malformations; * ordinal, ** dichotomous.

**Table 4 jcm-15-00372-t004:** Baseline characteristics of patients with and without propranolol treatment.

Characteristic	Without Propranolol (n = 107)	With Propranolol (n = 44)	*p* Value
Median age (IQR)	47 (34–62.5)	62 (48.7–68)	<0.001
Gender female, n (%)	68 (63.6)	27 (61.4)	0.946
HMO, n (%)	19 (17.8)	3 (6.8)	0.140
Argentine nationality, n (%)	102 (95.3)	43 (97.7)	0.820
**Vascular malformations, n (%)**			
CNS involvement	22 (20.6)	14 (31.8)	0.206
Hepatic involvement	72 (67.3)	38 (86.4)	0.028
Pulmonary involvement	47 (43.9)	17 (38.6)	0.677
Digestive involvement	13 (12.1)	12 (27.3)	0.042
Any visceral involvement ^#1^	92 (86)	39 (88.6)	0.862
**Hemorrhagic conditions or comorbidities, n (%) ^#2,^****			
Von Willebrand disease	1 (0.9)	1 (2.3)	0.999
Anticoagulant use *	3 (2.8)	3 (6.8)	0.491
**Specific treatments for HHT other than propranolol, n (%)**			
Antifibrinolytics	18 (16.8)	23 (52.3)	<0.001
Hormonal therapy or analogs	5 (4.7)	8 (18.2)	0.018
Bevacizumab	1 (0.9)	4 (9.1)	0.041
Other antiangiogenic agents ^#3^	1 (0.9)	4 (9.1)	0.041
At least one non-propranolol drug ^#4^	25 (23.4)	32 (72.7)	<0.001
**Compliance with nasal hygiene and lubrication measures, n (%) ***			
Insufficient	81 (75.7)	11 (25)	
Partial compliance	20 (18.7)	18 (40.9)	<0.001
Optimal compliance	6 (5.6)	15 (34.1)	
Non-compliance with nasal hygiene/lubrication ^#5,^**	81 (88)	11 (11.9)	0.001
**Baseline surgical or ablative treatments, n (%)**			
Major surgery	3 (2.8)	1 (2.3)	0.999
Young’s procedure	0	2 (4.5)	0.151
Minor surgery	10 (9.3)	7 (15.9)	0.381
**Iron support and/or transfusions, n (%)**			
Oral iron	24 (22.4)	18 (40.9)	0.035
Intravenous iron	17 (15.9)	11 (25)	0.281
Oral and/or intravenous iron	34 (31.8)	27 (61.4)	0.001
RBC transfusions	5 (4.7)	1 (2.3)	0.820
**Baseline hemoglobin**			
Mean hemoglobin (SD)	12.1 (2.2)	11.1 (2.1)	0.007
Anemia ^#6^ n (%)	51 (47.7)	30 (68.2)	0.034
**Sadick–Bergler scale** **Intensity * n (%)**			
1	47(43.9)	19 (43.2)	
2	43 (40.2)	17 (38.6)	0.941
3	17 (15.9)	8 (18.2)	
**Sadick–Bergler scale** **Frequency * n (%)**			
1	43 (40.2)	10 (22.7)	
2	39 (36.4)	16 (36.4)	0.048
3	25 (23.4)	18 (40.9)	
**Sadick–Bergler scale ** n (%)**			
Sadick–Bergler intensity ≥ 3	17 (15.9)	8 (18.2)	0.917
Sadick–Bergler frequency ≥ 3	25 (23.4)	18 (40.9)	0.049
Digestive bleeding baseline (%)	12 (25)	8 (25)	0.993
Digestive bleeding at 6 months, n (%)	9 (18.8)	8 (25)	0.516

^#1^, presence of any visceral involvement with vascular malformations (pulmonary and/or hepatic and/or digestive and/or other organs; ^#2^, there were no cases of other bleeding disorders, such as hemophilia or coagulopathies, and no patients were receiving antiplatelet therapy; ^#3^, antiangiogenic treatment other than bevacizumab; ^#4^, any of the following pharmacological treatments (one or more): bevacizumab, hormonal therapies, antifibrinolytics, and antiangiogenic agents other than bevacizumab; ^#5^, total non-compliance with daily nasal lubrication and hygiene measures; ^#6^, hemoglobin values < 12 mg/dL in women and <13 mg/dL in men. n, absolute frequency; %, relative frequency in percentage; IQR, interquartile range (P25–P75); SD, standard deviation; HMO, Health Maintenance Organization; CNS, central nervous system; RBC, red blood cell; * ordinal; ** dichotomous.

**Table 5 jcm-15-00372-t005:** Association between propranolol administration and improvement of epistaxis.

Epistaxis Improvement	Cr OR (CI 95%)	*p* Value	Adj OR (CI 95%)	*p* Value
Without propranolol	reference			
With propranolol	2.2 (0.9–5.6)	0.079	2.8 (0.9–8.6)	0.083

Cr OR, Crude odds ratio; 95% CI, 95% confidence interval; Adj OR, Adjusted odds ratio. Model adjusted for propranolol use, age, oral and/or intravenous iron supplementation, baseline minor surgery, nasal lubrication measures, any treatment other than propranolol, baseline Sadick–Bergler frequency ≥3, and propensity score using inverse probability of treatment weighting (IPTW). IPTW: Von Willebrand disease, Sadick–Bergler intensity ≥ 3, baseline major surgery, and anticoagulant therapy.

**Table 6 jcm-15-00372-t006:** Association between propranolol administration and improvement in epistaxis intensity and frequency.

Epistaxis Improvement	Cr OR (CI 95%)	*p* Value	Adj OR (CI 95%)	*p* Value
**Intensity**				
Without propranolol	reference			
With propranolol	1.5 (0.7–2.9)	0.286	1.9 (0.7–5.2)	0.181
**Frequency**				
Without propranolol	reference			
With propranolol	2.9 (1.4–5.9)	0.004	3.8 (1.3–11.2)	0.016

Cr OR, Crude odds ratio; 95% CI, 95% confidence interval; Adj OR, Adjusted odds ratio. Model adjusted for propranolol use, age, oral and/or intravenous iron supplementation, baseline minor surgery, nasal lubrication measures, any treatment other than propranolol, baseline Sadick–Bergler frequency ≥3, and propensity score using inverse probability of treatment weighting (IPTW). IPTW: Von Willebrand disease, Sadick–Bergler intensity >3, baseline major surgery, and anticoagulant therapy.

## Data Availability

The data underlying this study are not publicly available due to privacy or ethical restrictions, but can be obtained from the corresponding author upon reasonable request.

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
