# Peer review of "Propranolol Reduces Epistaxis in Hereditary Hemorrhagic Telangiectasia: A Large Retrospective Study"

_jcm, 2026, doi:10.3390/jcm15010372_

Round 1

Reviewer 1 Report

Comments and Suggestions for Authors

Major comments regarding the content

Sella et al provide a retrospective controlled observational study on the effect of oral propranolol therapy on epistaxis in HHT patients.

Citing the discussion, „Therapeutic strategies in HHT remain limited. Although in recent years relevant advances have been made in understanding its pathophysiological mechanisms, current management is mainly based on the repurposing of drugs approved for other indications that can modulate biological processes involved in the disease.” I fully agree.

Propranolol may a promosing candidate in this group, regarding its safety, inexpensibility and wide accessibility.

The introduction section is clear, with regard to the pharmacodynamics (vasoconstrictive and antiangiogenic effects) and approved indications of propranolol and the limited evidence of its effectivity in HHT.

Patient selection criteria (Figure 1) are clear. Baseline characteristics of the included and axcludes cohorts are properly introduced in Table 2 and the Supplementary Material.

Question 1: Do the authors have any explanation for the female predmoninance in both the propranolol and the control group? This should be discussed in the manuscript.

Question 2: Why could be the residency of the excluded patients from outside Buenos Aires with significantly bigger chance? This should be discussed later in the appropriate section.

Comment 1: Included patients are significantly older and have significantly more AVMs. These differences can be explained by the study design: non-definite HHT patients were excluded. Due to the age-related penetrance of HHT, older patients fulfil more diagnostic criteria. This should be detailed in the discussion.

The assessment of propranolol in HHT-related epistaxis is based on the Sadick-Bergler scale which is introduced in Table 1. This is a relatively simple score system with only two decisive aspects of epistaxis: intensity and frequency.

Question 3: Why the authors did not choose the more complex Epistaxis Severity Score (ESS) which is the gold-standard patient-reported outcome measure for evaluating nosebleed severity in HHT patients? Beyond the frequency and intensity, it takes other variables into account, too: bleeding time, medical attention, anemia and tranfusion requirements. The secundary outcomes investigated in the present manuscript contain these variables missing from the Sadick-Bergler scale, so I don’t think that the manuscripts should be re-written with the ESS as the primary outcome. However, the choice of the Sadick-Bergler scale instead of the ESS should be discussed in the manuscript.

The baseline and end-line characteristics of the propranolol and non-propranolol groups are comprehensive and well demonstrated in Table 3 and Figures 2-11.

The typical candidate for propranolol therapy is older (therefore anticoagulated with a slightly more chance), with HHT-specific polypragmasia, good or excellent compliance, more frequent epistaxis (more epistaxis-specific surgeries, anemia and more iron supplements as a result). This is obvious from Table 3.

Question 4: In the non-propranolol group, the adherence regarding nasal care and moisturizing/lubrication had significantly improved in the study peroid. What is the reason for this?

Propranolol is a potential candidate to manage epistaxis in HHT, as properly shown in Figures 6-9 and Tables 4 and 5.

The Discussion section is a good interpretation of study data and their comparison with the literature.

This manuscript is unequivocally worth to publish, but prior to this, the above questions should be answered in the modified version.

Minor comments regarding the content

  1. Introduction, lines 54-57: „Clinically, HHT is characterized by telangiectasias affecting the skin and mucosal surfaces, particularly in the nose, oral and gastrointestinal tract, as well as by visceral arteriovenous malformations (AVMs) involving the central nervous system, lungs, liver, and gastrointestinal tract.” HHT-related lesions in the gastroinestinal tract are telangiectases and not AVMs. I refer to Curacao criterion No. 3. Visceral lesions in the fundamental paper of Shovlin et al. (reference No. 7 in the manuscript).
  2. Figure 1: „w/o” - abbreviations should be detailed in the figure legend.
  3. Results, lines 207-210: „The study population corresponds to individuals over 50 years of age, who are more likely to undergo endoscopic studies, in order to avoid including patients without indications for endoscopic studies. Therefore, n corresponds to 48 for the non-propranolol group and 32 for the propranolol group.” This paragraph should be transposed to the "Results on Gastrointestinal Bleeding" section.
  4. Results, Figure 3: The figure is shifted to the right, so the intravenous iron coloumns at 6 months are not visible.
  5. Results, Figure 4: The same problem, the „optimal” coloumns at 6 months are not visible.
  6.  

Typing/grammatical errors

  1. Figure 1: „adultos” – adults
  2. Figure 1, legend: „measurementt” - measurement
  3. Table 3: „Insufficiente” – insufficient
  4. Results, lines 269-271: „The proportion of patients receiving bevacizumab therapy remained unchanged between baseline and 6-month follow-up. With no significant differences in either group (without propranolol p = 1.00; with propranolol p = 1.00).” These two sentenced should be joined, divided by a comma.
  5. Discussion, lines 440-442: „Dupuis-Girod and colleagues also did not observe modifications in hemoglobin levels in the randomized double-blind trial of topical timolol versus placebo [37].” - Dupuis-Girod and colleagues did not observe modifications in hemoglobin levels in the randomized double-blind trial of topical timolol versus placebo, either [37].

Author Response

Major comments regarding the content

Sella et al provide a retrospective controlled observational study on the effect of oral propranolol therapy on epistaxis in HHT patients.

Citing the discussion, „Therapeutic strategies in HHT remain limited. Although in recent years relevant advances have been made in understanding its pathophysiological mechanisms, current management is mainly based on the repurposing of drugs approved for other indications that can modulate biological processes involved in the disease.” I fully agree.

Propranolol may be a promising candidate in this group, regarding its safety, inexpensibility and wide accessibility.

The introduction section is clear, with regard to the pharmacodynamics (vasoconstrictive and antiangiogenic effects) and approved indications of propranolol and the limited evidence of its effectivity in HHT.

Patient selection criteria (Figure 1) are clear. Baseline characteristics of the included and excluded cohorts are properly introduced in Table 2 and the Supplementary Material.

We thank the Reviewer for these positive and encouraging comments on our manuscript.

Question 1: Do the authors have any explanation for the female predominance in both the propranolol and the control group? This should be discussed in the manuscript.

We thank the Reviewer for this relevant question. The predominance of women in both cohorts is consistent with what has been observed in many studies on HHT, including our own prevalence study. Although HHT is an autosomal dominant disease that equally affects men and women, we believe this difference could be due, at least in part, to consultation bias, as women tend to have greater access to healthcare systems. In addition, women tend to have milder epistaxis than men before menopause, likely due to the protective effect of female hormones. However, after menopause, epistaxis severity increases in women as hormone levels decline. Since most of the women included in our cohort were beyond reproductive age, this hormonal shift may also reasonably explain the female predominance observed.

To clarify this point, we have added this information to the Results section of the revised manuscript (see page 7, after table 3).

Question 2: Why could be the residency of the excluded patients from outside Buenos Aires with significantly bigger chance? This should be discussed later in the appropriate section.

We agree with the Reviewer that the residency of excluded patients outside Buenos Aires was associated with a significantly higher likelihood of exclusion. Considering that propranolol treatment is off-label and its use is limited in those areas, patients living closer to our center may be more likely to receive this indication, as their proximity facilitates follow-up. Furthermore, patients from the city of Buenos Aires tend to attend consultations more frequently, which enables better monitoring and therapeutic adjustments.

As suggested by the Reviewer, this concern has been clarified in item “2.2. Variables” (page 3), as follows:

“The residency of excluded patients outside Buenos Aires was associated with a significantly higher likelihood of exclusion. Considering that propranolol treatment is off-label and its use is limited in those areas, patients living closer to our center may be more likely to receive this indication, as their proximity facilitates follow-up. Furthermore, patients from the city of Buenos Aires tend to attend consultations more frequently, which enables better monitoring and therapeutic adjustments.”

Comment 1: Included patients are significantly older and have significantly more AVMs. These differences can be explained by the study design: non-definite HHT patients were excluded. Due to the age-related penetrance of HHT, older patients fulfil more diagnostic criteria. This should be detailed in the discussion.

We thank the Reviewer for this insightful comment, and as suggested it has been added to the Discussion section (page 19, second paragraph):

“Indeed, the included patients were significantly older and had a higher number of AVMs than the excluded population. These differences can be explained by the study design, as patients without a definite HHT diagnosis were excluded. Given the age-related penetrance of HHT, older individuals are more likely to fulfill a greater number of diagnostic criteria.”

The assessment of propranolol in HHT-related epistaxis is based on the Sadick-Bergler scale which is introduced in Table 1. This is a relatively simple score system with only two decisive aspects of epistaxis: intensity and frequency.

Question 3: Why the authors did not choose the more complex Epistaxis Severity Score (ESS) which is the gold-standard patient-reported outcome measure for evaluating nosebleed severity in HHT patients?

Beyond the frequency and intensity, it takes other variables into account, too: bleeding time, medical attention, anemia and transfusion requirements. The secondary outcomes investigated in the present manuscript contain these variables missing from the Sadick-Bergler scale, so I don’t think that the manuscripts should be re-written with the ESS as the primary outcome. However, the choice of the Sadick-Bergler scale instead of the ESS should be discussed in the manuscript.

We sincerely thank the Reviewer for this helpful remark. To clarify this concern, we have added three sentences to item “2.2. Variables” (page 4, after table 1):

“In designing this study, we evaluated epistaxis using both the Epistaxis Severity Score (ESS) and the Sadick–Bergler scale, as these are the tools currently used at our center. Unfortunately, at the start of the study, there were not enough patients with complete data for both scales (Sadick–Bergler and ESS). By contrast, all patients had Sadick–Bergler assessments; therefore, we decided to use this scale for the present study.”

The baseline and end-line characteristics of the propranolol and non-propranolol groups are comprehensive and well demonstrated in Table 3 and Figures 2-11.

The typical candidate for propranolol therapy is older (therefore anticoagulated with a slightly more chance), with HHT-specific polypragmasia, good or excellent compliance, more frequent epistaxis (more epistaxis-specific surgeries, anemia and more iron supplements as a result). This is obvious from Table 3.

Question 4: In the non-propranolol group, the adherence regarding nasal care and moisturizing/lubrication had significantly improved in the study period. What is the reason for this?

We sincerely thank the Reviewer for these helpful remarks. We believe that the increase in nasal lubrication and humidification in the non-propranolol group results from a supportive, educational, and positive reinforcement effect during consultations over time. This explanation has been added to the Results section (page 8):

“We also observed an increase in nasal lubrication and humidification in the non-propranolol group. This is likely attributable to the supportive counseling, education, and positive reinforcement provided during follow-up consultations.”

Propranolol is a potential candidate to manage epistaxis in HHT, as properly shown in Figures 6-9 and Tables 4 and 5. The Discussion section is a good interpretation of study data and their comparison with the literature. This manuscript is unequivocally worth to publish, but prior to this, the above questions should be answered in the modified version.

We sincerely thank the Reviewer for these positive and encouraging comments on our manuscript.

Minor comments regarding the content

-Introduction, lines 54-57: „Clinically, HHT is characterized by telangiectasias affecting the skin and mucosal surfaces, particularly in the nose, oral and gastrointestinal tract, as well as by visceral arteriovenous malformations (AVMs) involving the central nervous system, lungs, liver, and gastrointestinal tract.” HHT- related lesions in the gastrointestinal tract are telangiectases and not AVMs. I refer to Curacao criterion No.3. Visceral lesions in the fundamental paper of Shovlin et al. (reference No. 7 in the manuscript).

We thank the Reviewer for this comment. While the original 2000 publication (reference #7) describing the Curaçao criteria is very specific regarding visceral AVMs, several more recent studies include the gastrointestinal (GI) tract as an organ that can also harbor AVMs. In fact, the digestive tract may present not only telangiectasias, but also AVMs, which can be identified through CT angiography, conventional angiography, endoscopy, and even surgical specimen analyses. Based on both the updated literature and our own clinical experience, we respectfully consider the term “AVMs in the digestive tract” to be appropriate. We hope this clarifies the concern raised by the Reviewer.

-Figure 1: „w/o” - abbreviations should be detailed in the figure legend.

As suggested by the Reviewer, the abbreviations “w/o”, “pts”, and “y/o” have been spelled out in the figure legend. Please note that we have also slightly modified the actual figure and moved to the figure legend some of the text that was previously embedded within the original Figure 1.

-Results, lines 207-210: „The study population corresponds to individuals over 50 years of age, who are more likely to undergo endoscopic studies, in order to avoid including patients without indications for endoscopic studies. Therefore, n corresponds to 48 for the non-propranolol group and 32 for the propranolol group.” This paragraph should be transposed to the "Results on Gastrointestinal Bleeding" section.

Thank you for noting the misplace of this redundant information. As suggested by the Reviewer, this paragraph has been merged into the "Results on Gastrointestinal Bleeding" section (page 10) as follows:

“Gastrointestinal (GI) bleeding was assessed in patients aged >50 years, both with and without propranolol treatment. This measure was applied to minimize selection bias, as after this age patients are more likely to undergo gastrointestinal evaluation either through colorectal cancer screening protocols (which in our center almost always include upper endoscopy) or because GI bleeding due to HHT becomes more frequent beyond this age. Thus, the study population included, 48 patients corresponding to the non-propranolol group and 32 for the propranolol-treated group.”

-Results, Figure 3: The figure is shifted to the right, so the intravenous iron columns at 6 months are not visible.

Thank you for noting this error. The size of figure has been reduced, so it can now properly fit into the corresponding page.

-Results, Figure 4: The same problem, the „optimal” coloumns at 6 months are not visible.

Thank you for noting this error. The size of figure has been reduced, so it can now properly fit into the corresponding page.

-Typing/grammatical errors

Figure 1: „adultos” – adults

We apologize for this typo, which has now been corrected in the revised Figure 1.

-Figure 1, legend: „measurementt” – measurement

Thank you for noting this typo. Please note that we have moved to the figure legend some of the text that was previously embedded within the original Figure 1

-Table 3: „Insufficiente” – insufficient

We apologize for this typo, which has now been corrected in the revised manuscript.

-Results, lines 269-271: „The proportion of patients receiving bevacizumab therapy remained unchanged between baseline and 6-month follow-up. With no significant differences in either group (without propranolol p = 1.00; with propranolol p = 1.00).” These two sentenced should be joined, divided by a comma.

We thank the Reviewer for the suggestion and we have modified the text as indicated.

-Discussion, lines 440-442: „Dupuis-Girod and colleagues also did not observe modifications in hemoglobin levels in the randomized double-blind trial of topical timolol versus placebo [37].” - Dupuis-Girod and colleagues did not observe modifications in hemoglobin levels in the randomized double-blind trial of topical timolol versus placebo, either [37].

We thank the Reviewer for the suggestion. The text has been modified as indicated.

Reviewer 2 Report

Comments and Suggestions for Authors

This study adds some promising evidence to the discussion about propranolol's potential role in managing epistaxis in HHT. It provides a rationale for further investigation, particularly with randomized trials, to determine whether this treatment could become a mainstay in clinical practice. Some points need to be revised:

  • Age is different in the 2 groups. The authors must discuss this more.
  • Lines 291-293: "Significant changes were observed in both... with clinical improvement (figure 7)." What do the authors mean with this sentence?
  • Figure 11 needs further explanation. Improve figure legend.
  • Lines 516-517: "The strengths of the study include: it represents the largest cohort reported to date" I think that this must be report also in the abstract.
  • Supplementary material can be move to result section.

Author Response

This study adds some promising evidence to the discussion about propranolol's potential role in managing epistaxis in HHT. It provides a rationale for further investigation, particularly with randomized trials, to determine whether this treatment could become a mainstay in clinical practice.

We thank the Reviewer for these positive and encouraging comments on our manuscript.

Some points need to be revised:

-Age is different in the 2 groups. The authors must discuss this more.

We thank the Reviewer for the remark about the age of the 2 groups. The propranolol group had a higher median age compared with the control group (62 vs. 47 years). This difference is probably explained by the age-dependent progression of epistaxis in HHT, with older patients being more likely to seek medical care. This explanation has now been included in the revised manuscript (See Results section, page 7, after Table 3).

-Lines 291-293: "Significant changes were observed in both... with clinical improvement (figure 7)." What do the authors mean with this sentence?

Thank you for the observation. We agree that the original wording could be improved for clarity. We have therefore revised it accordingly, based on our interpretation of Figure 7.

-Figure 11 needs further explanation. Improve figure legend.

Thank you. We have made modifications to the legend of Figure 11, and we trust that the associated text is now presented more clearly.

-Lines 516-517: "The strengths of the study include: it represents the largest cohort reported to date" I think that this must be report also in the abstract.

We sincerely thank the Reviewer for this remark. We have now included the cohort information into the abstract.

-Supplementary material can be moved to result section.

We thank the Reviewer for the suggestion. The Supplementary Table 1 has been moved to the Result section (page 3) of the revised manuscript.

Round 2

Reviewer 2 Report

Comments and Suggestions for Authors

Good